# Cytoplasmic Increase in Hsp70 Protein: A Potential New Biomarker of Early Infiltration of Cutaneous Squamous Cell Carcinoma Arising from Actinic Keratosis

**DOI:** 10.3390/cancers12051151

**Published:** 2020-05-03

**Authors:** Montserrat Fernández-Guarino, José Javier Zamorano León, Antonio José López Farré, Maria Luisa González Morales, Ana Isabel Sánchez Adrada, José Barrio Garde, Jose Antonio Arias Navalon, Pedro Jaén Olasolo

**Affiliations:** 1Department of Dermatology, Hospital Universitario Ramón y Cajal, Universidad de Alcalá, Instituto de Investigación Sanitaria del Hospital Ramón y Cajal (Irycis) 1, 2003 Madrid, Spain; pedro@pjaen.com; 2School of Medicine and Health Public and Maternal and Child Heath, Universidad Complutense de Madrid, 28003 Madrid, Spain; josejzam@ucm.es; 3School of Medicine and Medicine Department, Universidad Complutense de Madrid, 28003 Madrid, Spain; antonio.lopez.farre@med.ucm.es; 4Hospital Central de la Cruz Roja, Universidad Alfonso X El Sabio, 2003 Madrid, Spain; mgonzalezmorales@salud.madrid.org (M.L.G.M.); anabelsanchezadrada@hotmail.com (A.I.S.A.); jbarrio@salud.madrid.org (J.B.G.); 5Universidad Alfonso X El Sabio, Vllanueva de la Cañada, 28091 Madrid, Spain; josari@uax.es

**Keywords:** actinic keratosis, cutaneous squamous cell carcinoma, cytoplasm, skin cancer, heat shock protein

## Abstract

Background: Cutaneous squamous skin cell carcinoma (SCC) is the second most frequent type of non-melanoma skin cancer and is the second leading cause of death by skin cancer in Caucasian populations. However, at present it is difficult to predict patients with poor SCC prognosis. Objective: To identify proteins with expression levels that could predict SCC infiltration in SCC arising from actinic keratosis (SCC-AK). Methods: A total of 20 biopsies from 20 different patients were studied; 10 were SCC-AK samples and 10 were taken from normal skin. Early infiltrated SCC-AK samples were selected based on histological examination, and to determine the expression of proteins, fresh skin samples were processed by two-dimensional electrophoresis. Results: The expression levels of three proteins, namely alpha hemoglobin and heat shock proteins 27 and 70 (Hsp27 and Hsp70, respectively) were significantly increased in SCC-AK samples with respect to normal control skin. However, only the expression level of Hsp70 protein positively correlated with the level of SCC-AK dermis infiltration. Immunohistological examination suggested that increased expression of Hsp70 proteins seemed to mainly occur in the cytoplasm of keratinocytes. The increased cytoplasmic Hsp70 expression in SCC-AK was confirmed by Western blot experiments. Conclusion: Cytoplasmic expression of Hsp70 could be a potential biomarker of early infiltration of SCC arising from AK.

## 1. Introduction

Actinic keratosis (AK) is a skin lesion associated with the cumulative effects of the sun. With AK there is a risk of progression towards cutaneous squamous cell carcinoma (SCC), also known as epidermoid carcinoma, which is a common form of non-melanoma skin cancer (NMSC) [1,2]. It is estimated that the risk of progression of an AK to invasive SCC is between 0.025% and 16% per year, with ultraviolet (UV) exposure being the major environmental risk factor for all skin cancers [3,4].

SCC is the second most common type of non-melanoma skin cancer. Indeed, SCC is the cause of the majority of NMSC deaths, with an incidence of 16 per 100,000 in central Europe, and 356 per 100,000 sun-exposed white men in the United States [5]. Middle-aged and elderly people are relatively more susceptible to this disease than younger people.

Several characteristics have been suggested to be associated with a poor prognosis of SCC, with most based on structural features of the SCC, including tumor thickness, location (ear), or the desmoplastic histological subtype. In this respect, the tumor–node–metastasis (TNM) staging system classifies SCC on the basis of SCC diameter, Broder’s histological degree of differentiation, and the invasion of extra-dermal structures [6,7]. However, at present it is difficult to predict which patients are at higher risk of SCC dermis infiltration, local recurrence, or metastasis. Moreover, the importance of tumor thickness as a prognostic factor has also been suggested, with three main risk categories [7,8]: less than 2 mm (no risk of dissemination), between 2.1 mm and 6.0 mm (low risk of dissemination, 4%), and greater than 6.0 mm (high risk of metastasis, 16%).

However, it is well-recognized that early diagnosis and outcome prediction are probably the most effective measures to prevent disease progression and improve the survival rate of SCC patients.

While the use of biomarkers has become incorporated into the standard of care for numerous malignancies, the application of biomarker studies to predict SCC development has not yet been clearly established. Clinicians cannot predict whether AK will progress to SCC, which treatment is better for prevention, if some therapies are contraindicated, or when field-directed therapies may induce SCC. There has been scarce research on the molecular mechanisms involved in SCC progression, but reported data have suggested that immune-compromised patients, such as organ transplant recipients, present a higher frequency of SCC and metastasis than the general population [9]. However, there are also doubts with regard to immunodeficiency, since the risk of developing SCC can vary depending upon the immunodeficiency type. Other works have focused on UV-induced mutations on different genes as the main mechanisms of SCC development from AKs. Differential expression analysis of gene biomarkers has provided very useful information about the molecular behavior of AK and invasive SCC, but a lack of tumor-specific biomarkers remains an ongoing challenge [10,11]. For example, mutation of p53 in SCC has been identified as an early event and is present in 90% of SCCs, but it has also been detected in normal sun-exposed skin [12,13]. Other works have suggested the participation of proteins in the molecular agents involved in cell differentiation, apoptosis, proliferation, migration, and cell adhesion.

It would be very interesting to be able to identify biomarkers of early infiltration of SCC arising from actinic keratosis (SCC-AK). New technological approaches to study several proteins in a single sample may be helpful for this proposal. A previous study using long-term archival formalin-fixed paraffin-embedded samples and proteomics demonstrated differences in the expression of proteins between AK and SCC and particularly in thioredoxin domain-containing 5 protein, a protein disulphide isomerase that has been reported to be upexpressed in several other cancers. However, the authors did not analyze changes related to the degree of infiltration of SCC and the skin phototype, and the histological SCC subtypes of the included patients were not specified. Moreover, from a methodological point of view, the efficiency of protein recovery may be modified and influenced by the fixation protocols, fixation time, and sample age [14,15].

Therefore, our aim was to try to identify proteins present in early SCC-AK infiltration in fresh skin samples.

## 2. Results

### 2.1. Clinical and Histological Features

The clinical features of the included SCC patients are shown in Appendix A. Six patients were male and four were females. Nine presented with phototype II and one with phototype I; all had received average sun exposure (Appendix A). Five of the SCCs were localized on the head (face/ear) and five on the back of the hand.

Six of the volunteers in the control group were male and four were females. The control volunteers were significantly younger than the SCC patients (*p* < 0.05) (Appendix A). 

All of the included control samples were obtained from non-photoexposed areas (axilla, genitals, scalp, and nape). Nine of the control volunteers presented with phototype II and one was of phototype I. 

All the SCC samples showed overlying AK (Appendix A), and almost all (9/10) also had an AK at the edge (Appendix A). Other histological characteristics such as tumor size and infiltration level are also shown in Appendix A. It is remarkable that dermis infiltration was very initial (Appendix A)

### 2.2. Two-Dimensional Electrophoresis Analysis

Two-dimensional electrophoresis (2-DE) spots were analyzed and identified on the basis of a previously published study showing the human skin map proteome [16]. The measured spots were chosen when spots were expressed in at least 65% of the 2-DE gels within each of the two groups (control and SCC samples). Densitometric analysis of proteins was performed, and they were classified according to their main functional characteristics as structural proteins, heat shock proteins, antioxidant proteins, tumor markers proteins, transport proteins, or transcription factor proteins.

There were no statistical differences between control and SCC samples in the levels of expression of structural proteins, which included actin, annexin I, two identified isoforms of annexin IV, two isoforms of annexin V, and two isoforms of cytokeratin (Appendix A). There were also no differences in the level of expression of proteins associated with antioxidant and transcriptional mechanisms (Appendix A).

The expression levels of alpha hemoglobin were significantly higher in SCC samples with respect to control samples (Appendix A). The expression levels of heat shock proteins 27 and 70 (Hsp27 and Hsp70, respectively) were also higher in SCC samples than in controls (Appendix A).

### 2.3. Correlations between the Protein Expression and the Histopathological Characteristics of SCC-AK

The relationship between the degree of infiltration of SCC-AK and the level of expression of the proteins with statistical differences between the control and SCC-AK samples were analyzed (Appendix A). There was no association between the degree of infiltration and the levels of expression of alpha hemoglobin, Hsp27, or Hsp70 (Appendix A). However, Spearman analysis revealed a positive correlation between the level of expression of Hsp70 and SCC-AK infiltration levels II and III (Appendix A). Statistical significance was lacking when a more advanced infiltration degree was correlated with the expression level of alpha hemoglobin, Hsp27, or Hsp70 proteins (Appendix A). Therefore, only early phases of dermis infiltration of squamous epithelial cells were associated with expression levels of Hsp70 (Appendix A).

### 2.4. Hsp70 Expression in SCC-AK 

Immunohistochemistry analysis demonstrated that healthy epidermal keratinocytes expressed Hsp70 protein at the cell nucleus (Figure 1). Slight Hsp70 expression was also observed in the cytoplasm of control keratinocytes (Figure 1). In SCC-AK samples with slight SCC infiltration (levels II and III), Hsp70 staining was greater than in control samples. Indeed, in these SCC-AK samples, Hsp70 protein was also localized in both the cytoplasm and in the nucleus, but in the cytoplasm in particular the expression level of Hsp70 protein was markedly greater than in that cytoplasm from control samples (Figure 1). This overexpression of Hsp70 protein was found in all six SCC samples showing slight SCC infiltration (+), and was classified as intense in 50% of the cases (+++).

The expression level of cytoplasmic Hsp70 protein in SCC-AK samples was also analyzed by Western blotting. As Figure 2 shows, a higher significant cytoplasmic expression of Hsp70 protein was observed in SCC-AK samples as compared with control samples (Figure 2). Cytoplasmic Hsp70 upexpression remained higher in the SCC-AK group than in the control group when age was used as a covariant.

## 3. Discussion

The present work shows, for the first time to our knowledge, an association between Hsp70 expression and the early infiltration of SCC-AK. 

A biomarker that can stratify the risk of transformation of AK into SCC has still not been identified. Furthermore, for field-directed therapies used to reduce this risk [16], the induction of carcinogenesis in the treated field has been demonstrated [17,18,19]. Therefore, dermatologists need to be able to select specific therapies and assess patient risk in order to provide personalized medical treatment.

The cautious selection of tumors and the patients studied is critical, as AK and SCC share biomarkers between them and with normal photoexposed skin [12,13]. Immunosuppression, chronic sun exposure, and fair skin are known risk factors for SCC [3,9,10]. All of the included patients with SCC had similar phototypes and patterns of sun exposure, and none were immunosuppressed [20,21].

The SCC samples included in this study were of low risk according to their differentiation degree, size (<2 cm), depth of infiltration (<2 mm), and histological features [6,7,8,22]. All appeared over an AK, were representative of the most frequent type of SCC in clinical practice, were of low-risk, and showed the first stage of an AK developing into an SCC [4,23,24,25,26]. These criteria were considered important for the identification of possible early markers of SCC-AK infiltration.

Patients in the control group were comparable in terms of phototype, sun exposure, and gender, but were younger than the SCC group (*p* > 0.005). The control skin was located in non-photoexposed areas (axilla, genitalia) to minimize molecular and histological changes caused by ultraviolet radiation [12,13].

The molecular data suggested differences between control skin and ASC-AK samples in the expression level of three proteins, namely alpha hemoglobin, Hsp27, and Hsp70. However, Spearman analysis suggested that only the expression of Hsp70 protein was associated with SCC infiltration. Interestingly this association was only significant at low infiltration levels (levels II and III) but not with a higher degree of infiltration (level IV). This suggests the specificity of Hsp70 upexpression for the identification of low SCC-AK infiltration.

Hsp27 and Hsp70 showed higher expression in SCC than in normal skin. Hsp27 and Hsp70 have a protective role in stress and develop a variety of functions, including regulating cell growth and differentiation. Interestingly, both proteins have been reported to play a role in chemotherapy resistance [27]. Indeed, Hsp27 was reported to be abundantly expressed in malignant cells and has been described as a prognostic factor and metastatic potential marker in gastric, colon, and esophageal cancers [28]. On the other hand, overexpression of Hsp70 has been associated with invasion and poor prognosis in patients with Oesophagus Squamous Cell Carcinoma (OSCC) and is consequently a promising target for treatment [29]. Alpha hemoglobin has also been shown to have significantly increased expression in SCC-AK. Expression of globin chains has been reported in a number of non-erythroid tissues, including neuronal cells, mesangial cells, macrophages, and hepatocytes, as well as in several types of epithelial cancers [30]. The functional significance of globin chain expression in the cells and tissues remains to be elucidated, but it has been suggested to provide oxygen storage in tissue cells. Interestingly, Zheng et al. suggested that beta-globin, by enhancing anchorage-independent lung and breast cancer cell survival, facilitates distant metastasis [31].

Interestingly, immunohistochemistry experiments supported the existence of higher expression of SCC-AK, with a particular increase in the cytoplasm. In this regard, Western blot experiments also demonstrated significantly higher expression of Hsp70 protein in cytoplasm from SCC-AK cells than in control keratinocytes. This higher cytoplasmic Hsp70 expression in SCC-AK cells with respect to control cells was independent of age, as was suggested by the fact that in the regression analysis cytoplasmic Hsp70 levels remained significantly increased in the SCC-AK group as compared to the control group after using age as a covariant.

Hsp70 has been involved in increased tumor growth and metastatic potential by inhibiting cell apoptosis. In other cell types, depletion or inhibition of Hsp70 frequently reduces the size of the tumors and can even cause their complete involution. In this respect, Hsp70 has the ability to interact with different cytoplasmatic proteins involved in cellular apoptosis regulation. Indeed, it has been described that cytosolic Hsp70 interacts with BAG-1, an apoptotic blocker protein, through binding to the anti-apoptotic protein Bcl-2 [32] Moreover, it was described that Hsp70 promoted mitogen-activated protein kinase (MAP) phosphorylation, inducing growth and cellular survival and blocking the pro-oncogen BAX in mitochondria, avoiding cellular apoptosis [33].

Previous works have demonstrated Hsp70 upexpression in non-skin SCC. Wang found a significant correlation between Hsp72 (a subgroup of Hsp70) and progression of esophageal SCC [34]. However, a study using paraffin-embedded biopsy specimens of SCC from various sites of oral and para-oral regions did not reveal changes in Hsp70 expression [35]. These different results may be related to specific characteristics of SCC, ranging from the sample type to the degree of SCC infiltration, which may promote distinct molecular behaviors.

### Study Limitations

The main limitation of the present study was most probably the reduced sample size. However, the results were very consistent, and their specificity is also supported by the fact that by Spearman associations a significant correlation with only one of the all studied proteins was found. In addition, the Hsp70 expression level was only associated with a specific degree of SCC infiltration. However, we are aware that additional studies with greater patient numbers are needed. 

Another study limitation was the age difference between control and AK-SCC patients. This is probably because control skin was extracted from the leftover skin of benign excisions, which are more frequent in younger patients. However, as previously mentioned, changes in cytoplasmic Hsp70 expression observed in AK-SCC samples with respect to controls were not age-dependent since significant differences remained after using age as covariant in the linear regression model.

## 4. Materials and Methods 

### 4.1. Collection of Skin Samples

Cutaneous samples (size <5 mm) of well-differentiated cutaneous squamous cell carcinoma from sun-exposed regions (head, neck, dorsal hands) were obtained from 10 patients during micrographic surgery. Ten normal specimens were also obtained from <5 mm punch biopsies from non-sun-exposed areas of 10 normal volunteers. The demographic data of all patients (age, gender, and phototype), sample locations, and scores of a sun exposure questionnaire validated for the Spanish population were collected [22]. Non-immunosuppressed patients with similar phototypes and patterns of sun exposure were included. Patients taking thiazide diuretics were excluded. In addition, patients with systemic inflammatory, infectious, or other oncological diseases, and/or those subjected to any surgical procedures within the last 6 months were excluded. The work was performed according to principles outlined in the Declaration of Helsinki. All the included individuals signed an informed consent form and the local institutional Ethics Committee approved the study (Internal register number: FCS/2015/3) 

### 4.2. Histological Examination

A routine hematoxylin–eosin histological study was performed in the biopsies. An expert dermatopathologist studied the tumors and only SCCs arising over an AK were selected. In the SCCs studied, the presence of an AK at the edge, size in centimeters, infiltration in dermis in micron, level of infiltration (Clark, 1–5), presence of foci of ulceration, adnexal involvement, and elastosis in dermis were described. 

### 4.3. Two-Dimensional Electrophoresis (2-DE) of Plasma Proteins, Image Acquisition, and Analysis

As previously reported in detail, 2-DE was performed using 250 g of total protein [36,37]

Samples were loaded on immobilized gradient IPG strips (18 cm, pH 4–7) and isoelectric focusing was performed using a Protean IEF cell system (Bio-Rad, Montreal, QC, Canada). In the second dimension, proteins were resolved on 10% SDS-PAGE gels using a Protean II XL System (Bio-Rad). As previously reported, the gels were then fixed, silver-stained for 30 min, and scanned in a UMAX POWERLOOK III scanner. One 2-DE gel was used for each sample, and all of the here-identified spots were expressed in at least 70% of the 2-DE gels. Densitometric analysis of the spots was performed using Quantity One 4.2.3. software (Bio-Rad). The densitometric intensity of each spot was evaluated after subtracting background staining of the corresponding gel.

### 4.4. Immunohistochemical Examination of Hsp70 in SCC

Histological preparations of 5 μm thickness using different samples from the SCC and control groups were assessed to study the presence of Hsp70. An internal and external control of the procedure was used (normal skin). An immunohistochemical technique was performed using the Hsp70 monoclonal antibody (Ab2787) tested for in vitro studies in humans following the manufacturer’s instructions (Diagnostic Master^®^, Vitro, Madrid, Spain). Two dermatopathologists assessed the subcellular localization of Hsp70 and quantified the intensity as mild (+), moderate (++), or high (+++) in all samples. The correlation between observers was also assessed.

### 4.5. Cytoplasmic Hsp70 Expression in Cytoplasm by Western Blot

The cytoplasmic expression level of the Hsp70 protein was analyzed by Western blot. For this purpose, biopsies were lysed and cytoplasmic fractions isolated following the instructions of a commercial kit (Mitochondrial Isolation Kit, Thermofisher, Rockford, IL, USA) The cytoplasmic fraction was solubilized in Laemmli buffer containing 2-mercaptoethanol. The proteins were separated in denaturing gels of polyacrylamide and 15% SDS. The same amount of protein (40 g/well) was loaded into each well. To detect the protein expression level, proteins in gels were transferred to nitrocellulose membranes (Immobiliun-P, MiliporeÒ, Darmstadt, Germany) and blocked for 1 hour at room temperature in a buffer containing 5% BSA. After blocking, membranes were incubated with monoclonal antibody against the Hsp70 protein (1:1000; ab2787, AbcamÒ, Abcam, Cambridge, UK) and then incubated with secondary antibody (horseradish peroxidase conjugated IgG anti-rabbit, Abcam) at 1: 2500 dilution. Proteins were detected by enhanced chemiluminescence (ECLÒ, Amersham Biosciences, Cambridge, UK) and evaluated by densitometry (Quantity OneÒ, Bio-Rad Laboratories, Chicago, IL, USA). Molecular weight markers (Sigma, St Louis, MI, USA) were used for molecular mass determination. In order to compare the protein expression of the different proteins with the expression of another constitutive protein, the expression of β-actin was also analyzed. For this, a parallel gel was run with the same samples and then the membrane transfer was incubated with a monoclonal anti-β-actin antibody (1:2000, Sigma-Aldrich, Saint Louis, MO, USA) and the respective secondary antibody (IgE anti-Rabbit, Sigma-Aldrich) at a 1: 2500 dilution.

### 4.6. Statistical Analysis

Results were expressed as mean ± SEM. To compare the levels of protein expression between the control groups, the non-parametric Mann–Whitney test was used. Correlation analyses between tumor depth and the expression of proteins were performed by the Rho–Spearmen test. To control the influence of age on Hsp70 expression, linear regression analysis was performed. The dependent variable was the level of Hsp70 expression, the independent variable was the experimental groups (Control, SCC-AK), and age was used as a covariant. The SPSS version 22.0 program (SPSS Inc., IBM, Chicago, IL, USA) was used. A value of *p* < 0.05 was considered statistically significant.

## 5. Conclusions

Slight dermis infiltration (level II and III) of SSC-AK was associated with higher Hsp70 levels in the biopsies. Hsp70 levels seem to be mainly increased in the cellular cytoplasm. This finding suggests Hsp70 as a potential early biomarker of SCC-AK infiltration, and therefore further studies are needed to confirm this finding.

## Figures and Tables

**Figure 1 cancers-12-01151-f001:**
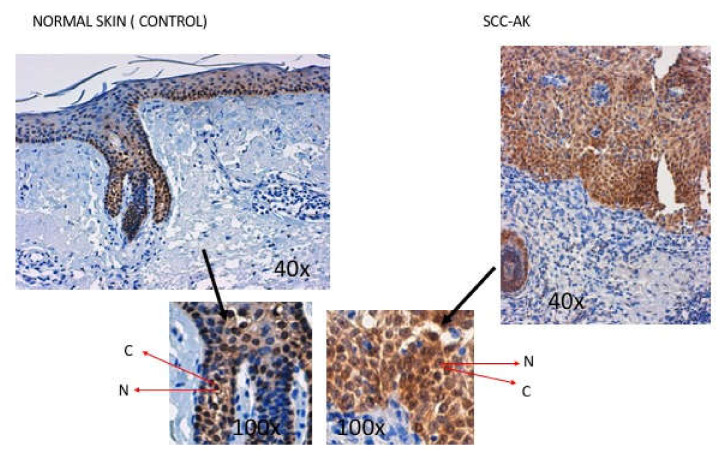
Immunohistochemistry analysis showing heat shock protein 70 (Hsp70) expression in control and squamous skin cell carcinoma arising from actinic keratosis (SCC-AK) samples with infiltration levels II and III. Abbreviations: nucleus (N) and cytoplasm (C).

**Figure 2 cancers-12-01151-f002:**
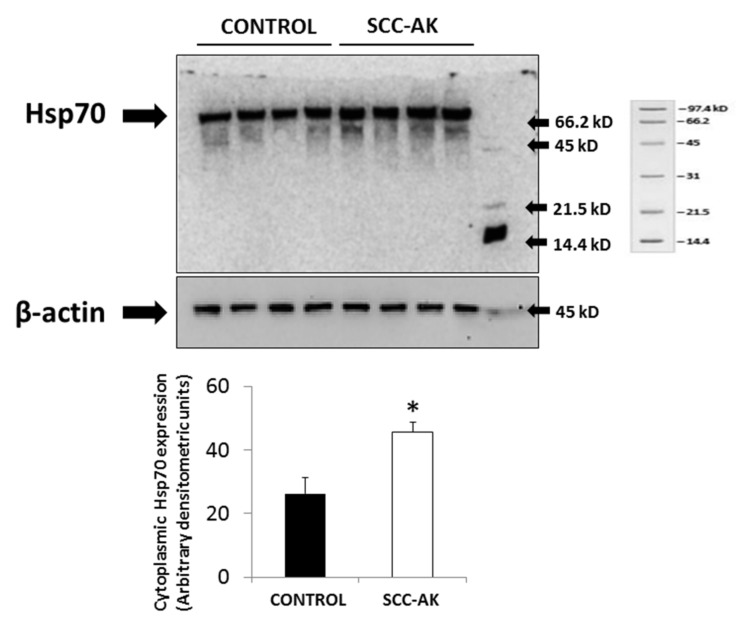
Representative Western blot experiments showing the expression of Hsp70 protein in cytoplasm from control and SCC-AK samples. β-actin was used as loading control. Bar graphs show the cytoplasmic expression of Hsp70 levels of all the Western blots. Results are represented as mean ± SD. * *p* < 0.05 with respect to control.

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
