# Peer review of "Cytoplasmic Increase in Hsp70 Protein: A Potential New Biomarker of Early Infiltration of Cutaneous Squamous Cell Carcinoma Arising from Actinic Keratosis"

_cancers, 2020, doi:10.3390/cancers12051151_

Round 1

Reviewer 1 Report

I prefer that Materials and methods are explained after introduction and not at the end of paper. 

In line 181 you put a double dots at the end of the paragraph.

in the next paper I'd like to know demoscopica features of SCC in order to identify, if it will be possible, some features which could help us to predict SCC invasion 

Reviewer 2 Report

In this manuscript, Fernández-Guarino and co workers examined the expression of Hsp70 in 20 AK and SCC samples and showed upregulation on AK/SCC. The major problem is that I cannot find tables 1 and 2 and 3 which show data pertaining to the samples and a summary of the 2D gel electrophoresis experiment from which they determined to look at the expression of Hsp70, and any evidence that there is a link between Hsp70 expression and the AK-SCC transition. I think there is a need to show the relevant 2D gel electrophoresis data and more relevant immunohistochemistry, and maybe show data in table 3 in a figure, as it is a significant part of the paper. 

The immunohistochemistry is poorly presented. I would like more examples of the staining, and if you are emphasising the nuclear and cytoplasmic stain, then some kind of analysis of prevalence of each in normal and SCC/AK would be a good idea. Furthermore magnifications are inappropriate, it would be better to use scale bars.

There are numerous typos and errors that need to be corrected throughout the manuscript.

Author Response

Please, see attachment

Reviewer 3 Report

Cancers

Manuscript ID: cancers-728790

This manuscript by Montserrat Fernández-Guarino and collaborators showed a correlation of slight dermis infiltration of SSC-AK, levels II and III, with increase expression of Hsp70 protein, suggesting Hsp70 as potential early biomarker of infiltration of squamous cell carcinoma arising from actinic keratosis. The research presented in the manuscript is interesting and novel. The procedures employed seem rational and the article appears well-designed and well-structured. However, the manuscript could be improved with some changes.

The authors demonstrated also that the expression levels of the alpha-hemoglobin and Hsp27 were significantly higher in SCC samples using two-dimensional (2-DE) electrophoresis. Moreover, this approach should be complemented with immunohistochemistry analysis.

The authors demonstrated alpha-hemoglobin, Hsp-27 and Hsp-70 were significantly increased in SCC-AK samples but only the expression of Hsp70 protein positively correlated with the slight dermis infiltration of SCC-AK. Could be interesting to conduce linear regression followed by ROC analyses and/or others statistical analyses in order to evaluate whether the signature of the three proteins was able to stratified SCC-AK patients.

Finally, I suggest adding a paragraph with information regarding Hsp27 and alpha-hemoglobin in SCC in discussion section.

Author Response

Please, see attachment

Round 2

Reviewer 2 Report

It is evident that you have made considerable improvements to this manuscript. I think it is good that the table are in the main body as they are critical to understanding the manuscript.    Major points  

Still quite a lot of typos, and the at least 10 references are missing, from the new parts of the discussion. This would need addressing

  The major issue, is that I would still like to see at least 1 example of staining of each grade of cancer from normal skin through AK to low and high grade SCC. Also the yellow bars are very difficult to see, I would prefer black ones if possible. You can remove the magnification form the legend if you had bars. What is the clear take home message. it seems to me that there is only a correlation between Hsp70 levels and infiltration in grade ii and iii tumours. Does this mean it has relevance in early stages of invasion? This needs to be expanded upon in the results and discussion.
  Minor points,    Table 2 it’s Annexin, not annexing.  Now that this is in effect multiple testing, has there been any corrections? HSP70 should still be statistically significant after multiple testing.   Table 3 is ambiguous. Do you mean nucleolar expression (which is very difficult to see in the displayed immunohistochemistry), or do you mean nucleus? - also cytoplasm. Also you don’t define the meaning of the cross scoring in the new parts the table.    The tables are no longer in the supplementary info, they are in the main body of the manuscript.

Round 3

Reviewer 2 Report

No further comments